# Alterations of Vaginal Microbiota and *Chlamydia trachomatis* as Crucial Co-Causative Factors in Cervical Cancer Genesis Procured by HPV

**DOI:** 10.3390/microorganisms11030662

**Published:** 2023-03-06

**Authors:** Ciro Gargiulo Isacco, Mario G. Balzanelli, Stefania Garzone, Mara Lorusso, Francesco Inchingolo, Kieu C. D. Nguyen, Luigi Santacroce, Adriana Mosca, Raffaele Del Prete

**Affiliations:** 1Department of Interdisciplinary Medicine, School of Medicine, University of Bari “Aldo Moro”, 70124 Bari, Italy; 2SET-118, Pre-Hospital and Emergency Department, SG Giuseppe Moscati Hospital, 74120 Taranto, Italy

**Keywords:** reactive oxygen species (ROS), human papilloma virus (HPV), *Chlamydia trachomatis*, vaginal microbiota, sexually transmitted infections (STIs), Cervix Cancer (CC)

## Abstract

*Chlamydia trachomatis* and human papillomavirus (HPV) are the most common pathogens found in sexually transmitted infections (STIs), and both are known to increase the risk of cervical cancer (CC) and infertility. HPV is extremely common worldwide, and scientists use it to distinguish between low-risk and high-risk genotypes. In addition, HPV transmission can occur via simple contact in the genital area. From 50 to 80% of sexually active individuals become infected with both *C. trachomatis* and HPV viruses during their lifetime, and up to 50% become infected with an HPV oncogenic genotype. The natural history of this coinfection is strongly conditioned by the balance between the host microbiome and immune condition and the infecting agent. Though the infection often regresses, it tends to persist throughout adult life asymptomatically and silently. The partnership between HPV and *C. trachomatis* is basically due to their similarities: common transmission routes, reciprocal advantages, and the same risk factors. *C. trachomatis* is a Gram-negative bacteria, similar to HPV, and an intracellular bacterium, which shows a unique biphasic development that helps the latter continue its steady progression into the host throughout the entire life. Indeed, depending on the individual’s immune condition, the *C. trachomatis* infection tends to migrate toward the upper genital tract and spread to the uterus, and the fallopian tubes open up a pathway to HPV invasion. In addition, most HPV and *C. trachomatis* infections related to the female genital tract are facilitated by the decay of the first line of defense in the vaginal environment, which is constituted by a healthy vaginal microbiome that is characterized by a net equilibrium of all its components. Thus, the aim of this paper was to highlight the complexity and fragility of the vaginal microenvironment and accentuate the fundamental role of all elements and systems involved, including the *Lactobacillus* strains (*Lactobacillus gasseri, Lactobacillus jensenii, Lactobacillus crispatus*) and the immune–endocrine system, in preserving it from oncogenic mutation. Therefore, age, diet, and genetic predisposition together with an unspecific, persistent low-grade inflammatory state were found to be implicated in a high frequency and severity grade of disease, potentially resulting in pre-cancerous and cancerous cervical lesions.

## 1. Introduction

Cervical cancer (CC) is one of the most frequent cancers with one of the highest rates of death among women worldwide. CC accounts for approximately 10% of the total newly diagnosed cancer cases and 8% of the total cancer deaths [1]. The development of CC is generally related to the presence of pathogens such as the human papillomavirus (HPV), which is the principal etiological agent [2]. CC is frequently diagnosed in women between the ages of 35 and 65 years (20% of cases of CC are found in women aged over 65 years), though it may silently commence at a very young age, even earlier than 20 years of age [1,2].

Nevertheless, the immune system is able to resolve the majority of HPV infections with only a small percentage progressing to precancer and cervical changes; the majority of patients remain asymptomatic or show transient positivity [3,4,5]. In fact, as confirmed by recent outcomes, the presence of only HPV is not sufficient to explain the development and progression of CC. Recent evidence shows that another agent needs to be present to increase the carcinogenic process, and *Chlamydia trachomatis* is the strongest co-infectious agent [3]. The data revealed that a higher prevalence of *C. trachomatis* infection was observed concomitantly with HPV-positive women (7–10%) than that in HPV-negative women (>4%) in the presence of CC [3].

*C. trachomatis* and HPV are common sexually transmitted infections (STIs) that share structural traits, and the DNA of both pathogens can be detected in approximately 99% of CC cases. However, whether both agents should be considered the ultimate risk for the insurgence of CC would not completely explain the whole picture, especially if one considers the great diversity in time manifestation. The doubts are related to the infection “roller-coaster” mode, which increases the need to investigate additional factors that could eventually be responsible for the process, such as the role of local microbiota and the immunity status. Intriguingly, this position was formerly confirmed by a few studies that tested the effects of both HPV and *C. trachomatis* infections on males diagnosed with chronic prostatitis in the presence of local dysbiosis [3,4,5,6]. The results obtained from urine specimens of both healthy males and controls revealed that the microbiota differed according to different urologic diseases, such as urinary incontinence, neurogenic bladder dysfunction, urologic chronic interstitial cystitis, and chronic nonbacterial prostatitis. Nowadays, we also know that alterations in the normal gut microbiome may affect the urinary tract, causing specific urologic diseases [7,8,9].

Interestingly, tissue alteration due to an unbalanced microbiota is not always related “de facto” to the presence of perilous microorganisms, as it was demonstrated it could be a consequence of the inappropriate use of antibiotics. The absence of any identifiable bacterial infection could be a distinctive feature of prostatitis, while an unnecessary use of antibiotics that alters the microbiota protective shield may be a precipitating factor, favoring the entry of opportunistic infectious agents [7,8].

This chicken-or-egg dilemma may add some confusion. Therefore, why can some individuals get through the infection without problems while others have difficulties in fighting it? Though there is a general agreement about immunity and hormones, little consensus remains on the role of the microbiome and altered microbiota with dominant *Lactobacillus iners* over *Lactobacillus crispatus* with a concomitant decrease in *Lactobacillus, Bifidobacterium, Enterobacter, Atopobium,* and *Streptococcus* [9].

Those changes lead to the consideration of carcinogenic mutation as a multistep process in which multiple factors are involved during a period of time. This scenario is characterized by an overlapping condition in which cells infected with pathogens are going through DNA alterations, and this enables genetic and epigenetic events that allow for viral replication and set the perfect stage for neoplastic changes [7,8,9,10,11,12,13].

These mutations take place concomitantly with changes in the expression of immune surveillance components and the system’s homeostasis that are shown to take place in response to oxidative stress. Important mutant transcriptomes are related to those genes that regulate scavengers in the enzymatic defense against oxidizing agents, which eventually damage proteins and nucleic acids, in particular, the expression of SOD2, CCP1, and CTT1 genes because they are involved during respiratory metabolism in limiting the production of high reactive oxygen species (ROS) [12,13].

Furthermore, an immune-compromised microenvironment, characterized by the presence of single-nucleotide polymorphisms (SNPs) that may affect the expression of immune-modulatory cytokines (interferon gamma-IFN-γ, tissue growth factor beta-TGF-β, interleukin 10 and 1β), should also be considered congruent of an altered microbiota condition, as previously described, characterized by dominant *Lactobacillus iners* over *Lactobacillus crispatus, Bifidobacterium, Enterobacter, Atopobium,* and *Streptococcus* [7,8,9,10].

However, the peculiar trait of end-stage cancer can only be possible via the persistent co-infection characterized by the concomitant presence of HPV and *C. trachomatis*. According to some scientists, there is a mutual benefit between the two; while *C. trachomatis* generates the convenient substratum that leads to the permeability of the surrounding tissues and allows HPV penetration into the epithelial cells, HPV alters the immune detection, which allows *C. trachomatis* to spread and multiply [3,4,5,14,15,16].

In the presented paper, we extended these findings by bringing together different aspects that are sequentially involved in the CC etiology. We proposed an all-inclusive perspective that includes the pathogens’ role, genetic make-up, microbiota integrity, and immune responses, which are all patterns we consider crucial in understanding CC pleiotropism. In this view, HPV and *C. trachomatis* are identified as the players responsible for triggering chronic and recurrent infections, which procure long-term inflammation of the genital area, alter the local immune mediator’s responses, increase the production of ROS and free radical generation, and cause the tissue and cell damage that promote local degenerative mutations (Figure 1 and Figure 2).

## 2. The Key Role of Local Microbiota

The vaginal microbiota is a very dynamic ecosystem, and similar to the gut, the local microbiota is composed of different strains with unique features and tissue interactions. The vaginal microenvironment is a densely populated area in which local bacteria perform a great variety of bioactivities under the guide of specific gene expression that is in charge of the enzymes necessary for specific defenses and biotransformation [17,18]. In fact, patients affected by both colorectal cancer and CC showed lower levels of butyrate-agent producers in the local microbiota than healthy individuals, indicating an important shift in terms of local flora homeostasis [17,18,19].

The vagina and ectocervix microenvironments are characterized by the presence of specific lactic acid-producing bacteria, which are capable of secreting a particular lubricant that traps invading pathogens [20]. The bacteria that make up the vaginal microbiome are inextricably linked to the acidity of the vaginal environment. The local flora is crucial for the local microbiome homeostasis, and the whole chain of bioreaction takes place with a specific mean pH of 4–4.5. A vagina with a pH of 4.5 or below serves as a protective shield against harmful pathogens that are unable to survive in such an acidic environment. When the pH is elevated (alkaline), harmful bacteria gain an opportunity to move in and disrupt the vaginal ecosystem [20,21,22].

Therefore, any changes in local balance homeostasis and in pH may have a direct, negative impact on the mucosa integrity that is functional to the outside and inside permeability gradient [21,22]. Once broken, the mucosa and tissues become an easy target of aggressive microorganisms and inflammatory processes that accelerate the deterioration process of both the endothelial wall and microbiota shield, enhancing the accumulation of pro-inflammatory endotoxins and long-term pathogen allocation (Figure 3) [20,21,22].

Results confirm that the degree of severity of cervical neoplasm is linked to these local and systemic changes that negatively affect the microbiota balance and functionality, especially against health-associated strains such as *Lactobacillus* spp., which leads to an increase in anaerobic bacteria, including *Gardnerella* spp., *Prevotella* spp., *Atopobium vaginae* and *Sneathia* spp., *Megasphaera* spp., and others [23]. Changes in the vaginal microbiota composition across cervical carcinogenesis may also lead to a profound switch to high pH with ROS elevation, which is considered a general marker indicative of inflammation, abnormal apoptosis, angiogenesis, hormonal imbalance, metabolic dysregulation, suppressed local antitumor immunity, and disease progression [23,24].

Both in vitro and in vivo models confirmed the effects of different *Lactobacilleae* strains on both *Chlamydia trachomatis* and HPV infection. It was found that *L. iners, L. crispatus, L. jensenii, Ligilactobacillus salivarius, L. gasseri, Limosigilactobacillus mucosae,* and *Limosigilactobacillus reuteri* all significantly reduced *C. trachomatis* infection in a dose- and time-dependent manner. The strongest anti-Chlamydia effects were linked to *L. crispatus* (90 percent reduction), whereas the poorest was found in *L. iners* (50 percent reduction). The D (–) isomer of lactic acid (LA) is a crucial component in *Lactobacillaceae* cell-free supernatants that are able to inactivate either *Chlamydia* Ebs or HPV, showing a positive correlation with anti-Chlamydia activity by neutralizing the pH value to 7.0 [24]. The outcomes from models that were inoculated intravaginally with *Lactobacillus* mixtures (*L. crispatus, Limosilactobacillus reuteri,* and *L. iners* at a ratio of 1:1:1) following genital *Chlamydia* infection confirmed this trend of decreasing the shedding activity in both the lower genital tract and the intestinal tract [24]. The beneficial effects are predominantly in the capacity modulation of super-reactive T cells, NK cells, and macrophages (M1), consequently reducing redounding cytokine production (TNF-α, IFN-γ, and IL-1β) in the vagina and diminishing overall genital tract inflammation and pathogenicity [25,26].

Both L and D isomers of LA are strictly dependent on strains with an active protonated form of LA that predominate at a pH below 3.9 [25]. Intriguingly, therapy using both isoforms of LA at pH 3.9 promoted better integrity of the barrier on ectocervical epithelial cells compared to untreated cells (Figure 1), results that were not achieved with media acidified to the same pH with HCl. Nevertheless, a genital environment dominated by *Lactobacillus* spp. is shown to keep the area pathogen-free and is associated with optimal pregnancy outcomes [25,26].

## 3. The Immune Profile of the CC Microenvironment *C. trachomatis* and HPV Crosstalk

In CC, a compromised microenvironment is mainly a consequence of deconstructed constituent parts of the vaginal microenvironment [26,27]. The intrinsic problems recall the simultaneous presence of multiple factors, such as the high inflammatory state, the presence of SNPs on genes that regulate the expression of immunosuppressive cytokines and interleukins, and a persistent local dysbiosis; these, in turn, facilitate the long-term aggressiveness of the HPV and *Chlamydia* infection [19,28]. The outcomes on the female genital tract, either on the vaginal or cervical microbiota, together with a substantial cytokine profile have identified the changes in microbiota diversity among women affected by HPV and *C. trachomatis* infection with bacteria/viral vaginosis (BVV) [28,29,30,31]. The co-infection with *C. trachomatis* and HPV (especially the 16, 18, 31, 33, 53, and 56 genotypes) is considered the most important risk factor for the presence of CC, especially in young, unmarried women who started their sex lives early with several sexual partners and who used oral contraceptives [14,15,16,22]. Interestingly, from these studies, the prevalence of *C. trachomatis* was high, predominantly in young women [14,15,16].

The immune response to acute HPV infections is initially mediated by mucosal NK cells, macrophages (M1 in the acute phase and M2 in the reparative phase), and epithelial cells, which produce antimicrobial peptides with known anti-viral/bactericidal effects. Since HPV has evolved molecular strategies to escape innate and adaptive immunity, inflammatory patterns are rarely observed. The infection is often marked with a rather high number of CD4+/CD25+ regulatory T cells and the presence of activated TH2 cells with the suppression of cytotoxic functions that lead to T cell anergy, a condition that may explain the increased rate of co-infective patterns due to other sexually transmitted disease pathogens such as *C. trachomatis* [28,32].

A few important HPV genotypes, such as the16 genotype, show the capacity to interact with *C. trachomatis* development and generate a sort of *C. trachomatis* steady persistence [31,32]. At this point, many authors describe the deep interchangeable behavior between the two pathogens; at transcriptional and post-translational levels, both HPV and *C. trachomatis* interfere with the reprogramming mechanisms of the host cells, which subverts the self-repair mechanisms [31,32]. This sabotage also involves the stem cell cloning mechanism, reducing the local regenerative capacity and leading to an increase in tissue damage [31,32].

Of note, the intracellular life of *C. trachomatis* is characterized by a vacuole surrounded by a membrane called “inclusion” in which it replicates followed by the transition from EB to RB and back to EB, the final form ready to exit the cell and infect other cells [3,4,5]. By this moment, *C. trachomatis* starts triggering the activation of the oncogenic pathway of Ras–Raf–MEK–ERK with the production of ROS to create the ideal microenvironment to support cancer cell growth [4,5]. The mechanism is mostly based on the ability of *C. trachomatis* to create mitotic spindle defects by aborting the spindle assembly checkpoint (SAC), causing the host cell to prematurely exit mitosis without the right corrections [30,31,32,33]. Nonetheless, *C. trachomatis* tends to subvert the host’s histones, triggering the upregulation of PH2AX and H3K9me3, both hallmarks of DNA double-stranded breaks (DSBs) and senescence-associated heterochromatin foci (SAHF) [30,31,32,33].

It is confirmed that supernumerary centrosomes have been identified in several types of carcinomas, as either addition or subtraction of chromosomes due to mitotic defects, and can reasonably be considered a sign of tumor growth and progression [30,31,32,33]. As previously mentioned, the increased ratio of ROS also contributes to DSBs that, in turn, elicit SAHF formation in an ERK-dependent manner. It should be mentioned that *C. trachomatis* is capable of blocking the use of DNA damage response (DDR) proteins, such as pATM and 53BP1, despite these cells conserving their proliferative ability, which is supported by oncogenic signals involving ERK, Cyclin E, and SAHF [33,34]. Furthermore, both HPV and *C. trachomatis* can also act at post-transcriptional and post-translational levels interfering with those genes controlled by an E2F transcription factor and associated with the DNA mismatch repair mechanism [30,31,32,33,34].

Therefore, we assume that the possible correlation between *C. trachomatis* and HPV coinfection with cervical tissue changes could be based on the steady though active presence of both pathogens that pervasively and silently keep subverting the host’s detection and repairing mechanisms at a very molecular level, which explains the reason why a large percentage of affected women remain asymptomatic over a long period of time [6,34,35].

As a predictive tool of possible CC high risk among those females who are asymptomatic with tubal infertility, assessing the vaginal microbiota condition and the local immune expression in the presence of both HPV and *C. trachomatis* could be helpful. As a matter of fact, no significant differences in the phylum, class, and operational taxonomic unit (OTU) levels were observed among women with tubal infertility who were *C. trachomatis* negative and healthy controls [34].

## 4. The Crosstalk between Microbiota and the Immune–Endocrine System

Similar to the intestines, the vaginal microbiome composition is not equally expressed over the female reproductive tract. *Lactobacillus* spp. appears to be the majority in the uterus; the non*-Lactobacillus* spp. are major parts of the uterine cervix, while the vagina is normally predominated by *Lactobacillus* spp. of the Community State Types (CST) I, II, III, and V. The CST-IV is composed of a polymicrobial mixture of anaerobes, suggesting that any predominance of CST-IV is clinically related to bacterial vaginosis [36]. It is well accepted that innate immune responses are driven by the conditions of the vaginal bacterial community, and the CST-IV plays a potential role in scattering the higher pro-inflammatory responses compared to others [36].

Not only does the vaginal microbiome produce hormones, but it can also communicate to the endocrine glands how much of each hormone is needed. However, this relationship is bidirectional, since the host–bacterium interaction also depends on sex and hormones via interactions among its metabolites, the immune system, chronic inflammation, and nerve–endocrine/paracrine pathways [37]. Data show that early childhood microbial exposures play a key role in setting up sex hormone levels that, in turn, determine the typology of immune responses. Several attempts at microbiota transplantation in animal models from adult male mice to immature female mice led to a hormone level modification with an increase in testosterone level and metabolomic changes in autoantibody production [37].

For instance, the results from a study by Yurkovetskiy and colleagues showed that sequenced bacterial DNA extracted from the caecal contents of prepubescent mice (4 weeks old) and postpubescent mice (10–13 weeks old) did not have significantly differences, though the α-diversity between the two sexes in the prepubescent mice was observed. The 16S rRNA genes from the microbiota of males, females, and castrated males were sequenced, and the outcomes confirmed something “unusual”; the microbiome of the females was highly similar to that obtained from the castrated males, reflecting a similar composition and similar levels of androgen hormones [38].

In the same way, samples obtained during the pregnancy period showed that vaginal microbiota composition tends to modify, reflecting the changes in hormone level and typology. An increase in the phylum *Firmicutes* was observed from Trimester I to Trimester III, which characterizes the vaginal microbiota of healthy women [39]. Conversely, dysbiosis of the vaginal microbiota during BV has been reported to be characterized by diverse bacterial taxa that belong to specific phyla, such as *Fusobacteriota, Actinomytetota, Pseudomonodota,* and *Bacteroidota* [40]. Similarly, the vaginal microbiota of pre- and peri-menopausal women are mainly composed of the phylum *Firmicutes*, and the vaginal microbiota in post-menopausal women is confirmed to harbor mainly the phyla *Pseudomonadota*, *Bacteroidota,* and *Actinomycetota* [39,40].

However, a key fact is that there is a deep relation between the different components of the genital area based on the presence of specific receptors. Recognition cell receptors are located everywhere, either on squamous epithelial cells of the vagina or on the columnar cells lining the upper female genital tract, that are capable of recognizing the bacterial products expressed by the vaginal microbiome [41,42]. Of course, the vagina, endometrium, and tubes are equipped with high-functioning immune cells and androgen receptors, which completely synchronize with the microbial environment. The interplay between the immune system and the microbiome involves hormone receptors together with different immune mediators; stem cells; leukocyte subsets; plasma cells; IgG, IgM, and IgA antibodies; interleukins; and inflammatory proteins [41,42]. Lactobacilli play a pivotal role in modulating both inflammatory and anti-inflammatory responses and directly modulating the local production of interleukins IL-1β, IL-6, and IL-8, and IL-2, IL-10, and IL-173 [41,42] (Figure 4).

Immunologically, hormones play a fundamental role as they actively participate in regulating the production of antimicrobial peptides (β α-defensins, SLPI) and pro-inflammatory cytokines that are expressed by local epithelial cells, especially in ensuring the safety of sperm and preventing infections [36]. Estrogens are the major contributors to microbial population selection during the different aging phases; for instance, in prepubertal age, the vaginal microbiota is mainly characterized by anaerobic species of the *Enterobacteriaceae* and/or *Staphylococcaceae* family, and in puberty, the estrogens promote the accumulation of glycogen by mature epithelial cells [36,37,38]. The presence of both maltotriose and alpha-dextrins, derived from glycogen digestion, is an extremely important food for *Lactobacillus* species and their production of LA [37,38].

The microbiota system, and immune and endocrine systems are part of a unique hyper-specialized unit that ensures vital functions in the body and organizes and performs crucial activities. The microbiota and immune system cooperate to provide protection from lethal pathogens, whereas the microbiota and endocrine system ensure the proper metabolic function of peripheral organs by regulating systemic homeostasis. In this view, the cross-talk between the microbiota, hormones, and immune system cells could be considered a further point in understanding vaginal infections and aggressive pathogens that may lead to carcinogenesis [41,42,43,44,45,46].

## 5. Modern Diet and *Lactobacillus* Issues

The balance of the vaginal microbiome is constantly influenced by various local and systemic factors, such as diet, hormonal levels, smoking, and the use of topical products or antibiotics. Aging plays a significant role in the composition of the vaginal microbiota, and vaginal physiology is modified by the composition of the vaginal microbiota [45,46,47]. During pregnancy, the first microbial colonization occurs at the time of the delivery, supplied by the mother’s vagina or skin, depending on the route of birth. In female newborns, both the vulva and vagina are affected by the presence of transplacental estrogenic residues, which provides a glycogen supply that is metabolized by endogenous bacteria and lowers the vaginal pH [48]. As estrogens are metabolized, the vaginal glycogen content is lost, switching towards an alkalized pH that is neutral up to puberty. During puberty, the maturation of the adrenal glands and gonads induces a rise in the levels of estrogen as well as in the intracellular compartment. Two main types of colonies are determined at this stage: the *Lactobacillus* spp. and *Atopobium* and *Streptococcus* spp. [48].

However, age-related microbiota changes (dysbiosis) contribute to inflammation because long-term stimulation causes immune-senescence, rendering the host more sensitive to potentially harmful bacteria that, in turn, contributes to the higher rate of different pathological conditions in the elderly [38,39,40].

According to the dietary outlook in this small review, we can refer to “Healthy diet style” and “Unhealthy diet style”. The modern industrialized “Unhealthy diet” habits are defined by high loading of sugar, solid oils, highly refined carbohydrates and refined grains, fried potatoes, and sweet drinks and are substantially associated with higher BVV odds [44,45,46,47,48,49]. For instance, overweight women are more prone to have higher rates of infections, either bacterial or viral, compared to lean women due to different mechanisms, including alterations in hormonal, metabolic, or immunological functions that tend to affect the overall structural stability of the vaginal microbiota. For instance, an altered condition of the composition of the genital fluids, high glycogen levels, was observed following a high starch intake [40,41].

A higher risk of molecular BVV was observed in women with low vitamins D, A, E, β-carotene, and iron; the results were assessed by using 16S rRNA gene amplicon sequencing [42,43]. Interestingly, it has been proposed that the impact of certain micronutrients on the vaginal microbiota could be mediated through the effects on the gut microbiota. Of note, several studies have noted concordance between rectal and vaginal carriage of specific bacteria, including *Lactobacillus* spp. with a decreased risk of BVV [42,43,49]. Experiments performed on rats and pigs assessed via qPCR showed that supplementation with high levels of vitamins D, A, E, β-carotene, and iron improved the function of digestive enzymes and increased the relative abundance of the genus *Lactobacillus* in the gut microbiota [40,41,42,43,49]. Among these micronutrients, vitamin D attracted great attention. Outcomes reported the serum level of 25-hydroxy-vitamin D [25(OH)D] as negatively correlated with BVV in pregnant women during the first trimester, validated by different studies that associated the low level of vitamin D and BVV during pregnancy probably via the existing correlation between *Lactobacillus crispatus* and serum 25(OH)D [45,46]. However, it is unclear how applicable animal data may be to humans because the use of enteric bacteria can deconjugate estrogens and promote their reabsorption to the circulatory system, leading to glycogen and mucus production and thickening of the epithelium of the lower genital tract. Thus, a reduction in estrogen-metabolizing bacteria could influence the *Lactobacillus* dominance in the vaginal flora [45,46].

In addition, a high carbohydrate and fat intake induces dysbiosis via the down-expression of angiopoietin-like protein 4 (Angptl4), resulting in lipoprotein lipase (LPL) hyperactivity. The elevated LPL has direct consequences on higher uptake of fatty acids that, in turn, increase fat accumulation in peripheral tissues and thus inflammation [47].

*Bifidobacterium* spp., *Lactobacillus* spp., and *Prevotella* spp. are known to be highly sensitive to fat and refined sugars, which affects their role in the activation and modulation of the endocannabinoid system. These changes contribute to altering the gut microbial composition, generating the phenomenon known as “The leaky gut”, and allowing bacteria translocation within and in and out of the gut compartments. Furthermore, this typology of food has a direct impact on the overgrowth and migration of Gram-negative pathogens with bacterial fragments, such as lipopolysaccharides (LPS), freely moving across the intestinal lumen and triggering the nuclear factor kappa B (NF-κB) pathway in the bloodstream that, in turn, opens up the Toll-like receptors towards the activation of proinflammatory cytokine CD14, causing an increased intestinal permeability [47].

Smokers have a predisposition towards infection by *C. trachomatis* and HPV and CC insurgences. Data analysis found two pieces of evidence: (i) it was significantly more likely that smokers’ vaginal microbiota had a low *Lactobacillus* prevalence, and (ii) metabolites produced during heat combustion were increased in higher Nugent scores [47]. Smokers and non-smokers were studied and compared, and the results showed differences in vaginal metabolites. Among these women, biogenic amines were higher in smokers; these amines were shown to have some sort of effect on the virulence of infectious pathogens, contributing to vaginal malodor [50,51].

Stress also has the potential of threatening the balance and homeostasis of a vagina’s internal background [25]. Trauma and negative emotions unfavorably affect the individual’s sense of stability and depress immune responses; outcomes from animal models report that persistent exposure to psychosocial stress can lead to an encouragement of the hypothalamic–pituitary–adrenal–medullary axes (HPA), which drives a cortisol-induced inhibition of glycogen deposition in the vagina, ending with an interruption of epithelial maturation crucial to keep vaginal homeostasis [25]. This is a phenomenon particularly important during pregnancy, in which the high local production of corticotropin-releasing hormone occurs in the decidua, fetal membranes, and placenta, tending to compromise the fragile equilibrium between estrogen and lactobacilli activity and vaginal homeostasis [48,51,52,53].

## 6. New Strategies and Approaches in Clinic

As far as we know, intestinal and vaginal tissues share many features both physiologically and pathologically, including an immune system to protect from pathogen invasion and maintain homeostasis [53,54,55,56]. The great diversity of microorganisms composed of bacterial, archaeal, eukaryotic, and the dominant lactobacilli in the vaginal tract are strongly influenced by several factors, including host genetic make-up, host bionomy, and environmental factors [53,54,55,56].

Though the probiotic approach shows some positive results so far, results still remain uncertain with nothing particularly definitive about its efficacy; therefore, new strategies have been adopted that show new potential [57,58]. The idea was simple and adopted the basic concept of organ transplant using the fluid from a healthy vagina and transplanting it into an unhealthy one. In this case, it is all about the bacteria, since each individual carries practically trillions of bacteria living on and in their body, different strains located in different locations performing different activities [55,56,57].

In the vagina, the complexity follows a precise homeostasis regulated by the presence of *Lactobacillus crispatus* and hormones, such as estrogen, progesterone, and testosterone. The limitation of exogenous probiotic therapy is probably due to the fact that *Lactobacillus crispatus* and kindreds are not the same lactobacilli in dairy products nor the ones found in the gut [54,55]. The outcomes from a genomic study performed on *Lactobacillus crispatus* revealed that 37 strains had genes encoding 33 families of glycoside hydrolases (GHs). The levels of GH1, GH13_18, GH2, GH20, GH25, GH3, GH36, GH43_14, GH73, GH78, and GH92 differed significantly (*p* < 0.05) between those obtained and analyzed from the gut and those from the vagina. The *Lactobacillus crispatus* strains of gut origin contained more types of carbohydrate enzyme than the same strain from the vagina [54,55].

In addition, with a lower pH compared to the gut, which helps to reduce the risk of gynecological infectious diseases and maintain local health and homeostasis, the local *Lactobacillus crispatus* evolved to face the harshness of acidity [49]. The vaginal *Lactobacillus crispatus* expresses the gene that encodes the manganese transport protein, which takes up Mn^2+^ and, by this mechanism, expels protons out of the cell thus participating in the acid response and maintaining intracellular pH homeostasis. The Mn^2+^ is also an important tool that helps in protecting *Lactobacillus crispatus* against oxidative stress [54,55,56].

Therefore, under a similar genetic background, the possibility of microbiome transplantation (VMT) gives a valid alternative model that is based on the role of the entire microbiota in the host defense mechanism [58]. At the moment, there are several exploratory studies testing the use of VMT from healthy donors as a new alternative therapy for patients suffering from symptomatic, incurable recurrent bacterial/viral vaginal infection. The concept follows the positive results obtained by fecal microbiota transplantation (FMT) for recurrent infections due to destructive bacteria and the long-term inflammatory state (Clostridioides difficile and ulcerative colitis) [56,58].

However, the VMT varies and is not a one-size-fits-all procedure. The application may differ from person to person with a single procedure treatment to multiple top-up transplants. This indicates that compatibility requires the consideration of variables such as the species in the donor sample, the recipient’s baseline vaginal environment, or both, in order to reach a proper match for the treatment to be effective and to ensure that no pathogens of foreign bodies (sperm) could be transferred to the recipient. Sequencing the donor’s and recipient’s genetic patterns and microbes throughout will help to better assess the types of organisms that can establish themselves and thrive in each recipient, how the different communities vary based on the recipient’s vaginal status, and the specific traits of dysbiosis and local pathological condition such as endometriosis [58,59,60].

The therapeutic effects found with VMT show an almost consistent inhibitory effect on both inflammation (lowering cytokines and downregulating the NF-kB pathway) and ectopic lesion progression in mice, indicating that vaginal bacteria may play an important role in endometriotic lesion progression and reconstructing the vaginal microbiota microenvironment to normal [58].

In conclusion, the overall results confirm that the vaginal microbiota works as an antipathogen solid screen, reducing the progression of inflammation and endometriosis and eventually limiting carcinogenesis progression. A healthy microbiota and, eventually, the VMT approach have shown great benefits in the treatment strategy for several gynecological diseases [58,59,60].

## 7. Conclusions

The risk of CC is a multifactorial matter that involves different factors and variants. The HPV-*C. trachomatis* DNA–host integration is usually a necessary event in the pathogenesis of HPV-*C. trachomatis* related cancer; however, the mechanism of reciprocal validation and integration probably needs years and takes place in specific microenvironment conditions. In addition, one must consider that any breaks or damage in the local immune surveillance must also be facilitated by either genetic or epigenetic factors for the integration to occur. In this regard, studies show that viral–bacterial integration is indeed eased by local dysbiosis, which in turn increases the pathogens’ indiscriminate invasion and persistent inflammatory state. Furthermore, aging (hormone imbalances) and diet also have detrimental effects and contribute to inflammation and uncontrolled ROS persistence. Together, these heterogeneous factors lead to DNA strand breaks in the host cells that enable a stronger HPV-*C. trachomatis* reciprocal integration. Consistent with this assumption, the silent long-term HPV-*C. trachomatis* co-infections become the essential contributors to tissue transformation, which leads to CC. The prevalence of co-infection of *C. trachomatis* and HPV is, therefore, an important contributory factor that may begin at a young age and silently persist as long as the vaginal microbiota, immunity, and hormones keep preserving local homeostasis. Obviously, microbiological screening and the vaginal test still remain the most effective diagnostic tools. Meanwhile, the future of microbiota-based therapeutics shall be performed on well-defined groups of microorganisms, characterizing their specific beneficial effects on the host and the recipients. Therefore, precise identification of the microbial community members in either healthy individuals or in those with dysbiosis should be the strong theoretical foundation of future prospective research that needs to focus on the interactions between the microbiome and the host’s immunity as well as the interactions between age, sex, the genetic predisposition, and the underlying daily habits and diet behavior in both healthy individuals and patients. This approach will certainly raise the rate of protection against pelvic inflammatory disease and infertility, and potentially will be of great help in preventing and reducing the incidence of CC.

## Figures and Tables

**Figure 1 microorganisms-11-00662-f001:**
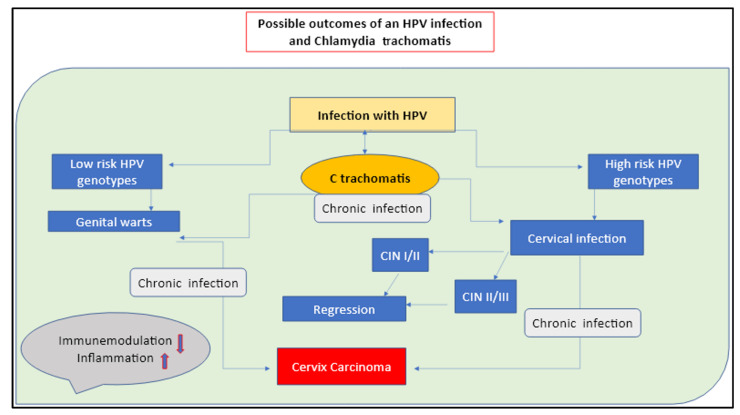
Possible outcomes of an HPV infection with a *C. trachomatis* co-infection. The long-term silent infection appears to be the key factor of CC insurgency.

**Figure 2 microorganisms-11-00662-f002:**
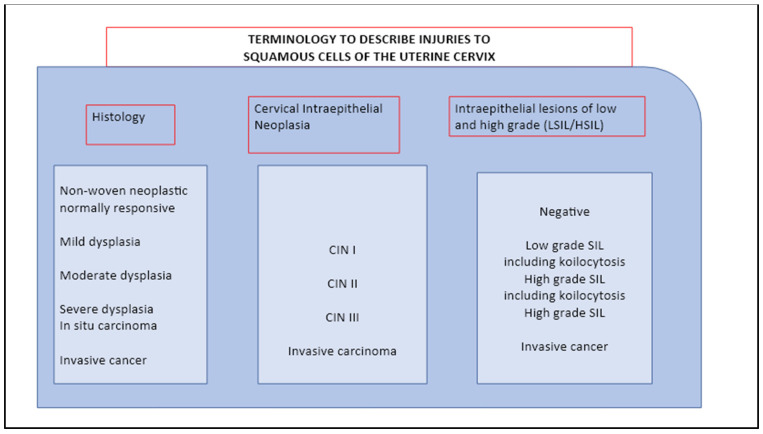
Epidemiology of HPV infection and cervical cancer.

**Figure 3 microorganisms-11-00662-f003:**
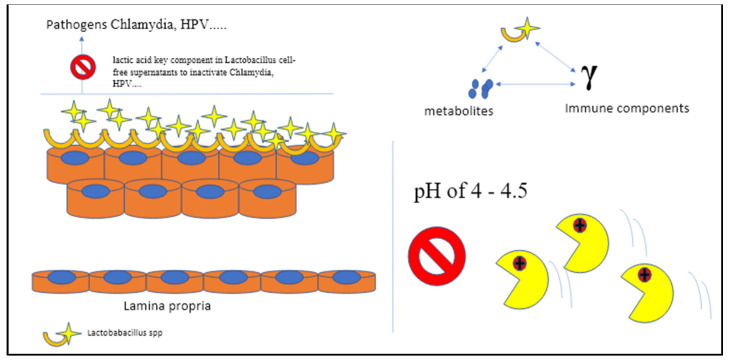
The bacteria that make up the vaginal microbiome are inextricably linked to the acidity of the vaginal environment. The local flora is crucial for the local microbiome homeostasis, and the whole chain of bioreaction takes place in spaces with a specific mean pH of 4–4.5. A vagina with a pH of 4.5 or below defends against harmful pathogens that are unable to survive in such an acidic environment (C Gargiulo Isacco, K CD Nguyen).

**Figure 4 microorganisms-11-00662-f004:**
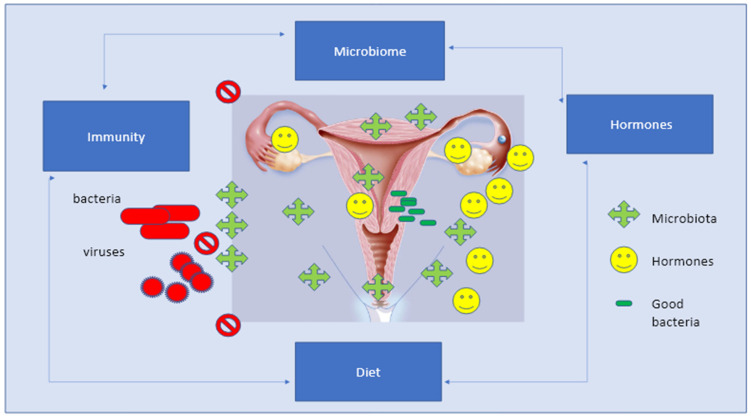
The Eubiosis of the female sex apparatus is based on the balance of several factors composed of microbiota, hormones, and local bacteria, which is then associated to regular and equilibrated immune responses, diet, and active life-style.

## Data Availability

This is a review type article, nevertheless part of data presented in this study are available upon request from the corresponding author at DIM Section of Microbiology and Virology of University of Medicine Aldo Moro Bari.

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
