# Peer review of "Alterations of Vaginal Microbiota and *Chlamydia trachomatis* as Crucial Co-Causative Factors in Cervical Cancer Genesis Procured by HPV"

_microorganisms, 2023, doi:10.3390/microorganisms11030662_

Round 1

Reviewer 1 Report

P2L59 - this sentence is confused because it seem to mean that C,trachomatis is a high risk factor for cervical cancer development. In addition, there is no reference here.
P8L285 - bacterial/viral vaginosis (BVV) is a confused term and also there is no reference

P2L47  - nearly 10% of the total newly diagnosed cancer cases 46 and 8% of the total cancer deaths  - there is no appropriate reference.

P2L55 - The worldwide outcomes generally agree that top three C.trachomatis genotypes were The 56

most common genotypes were E, followed by J, F, D, K, G, H, B, Ia. – there is no reference

P3L90  - SNPs and low levels of immune-modulatory cytokines such as interferon gamma (IFN-γ), tissue

growth factor beta (TGF-β) and interleukin 10 and 1β (IL-10, IL- 91 1β) – there is no appropriate reference.

P4L122 - Patients with both colorectal cancer and CC have showed lower levels of butyrate producers in the local microbiota than healthy individuals indicating an important shift in terms o local flora homeostasis [13-15]. –references 13-15 are not appropriate

P5L151  - Changes in the vaginal microbiota composition across cervical carcinogenesis may also lead to a profound switch into high pH with ROS elevation, considered general markers indicative of inflammation, abnormal apoptosis, angiogenesis, hormonal imbalance, metabolic dysregulation and suggestive of suppressed local antitumor immunity and disease progression –reference 19 is not appropriate

P6L180 - The immanent problems are to be related to the simultaneously presence of multiple factors such as, high inflammatory state, the presence of single nucleotide polymorphisms (SNPs) on genes regulating the expression of immunosuppressive cytokines and interleukins like TGF- ß1, IL-10 and IFN-γ which favor local dysbiosis that in turn facilitate the persistence of the HPV and Chlamydia infection – there is no reference

P6L188 - The vaginal microbial ecosystem and the cytokine profile play a role in promoting 188 cervical dysplasia, given that an abnormal vaginal microbiota has been associated with 189 the acquisition of long term C trachomatis and HPV infection [25,26]. - references 25-26 are not appropriate

P7L268 - In this scenario hormones are key actors as they are involved in regulating the production of antimicrobial peptides (β α-defensins, SLPI) and of pro-inflammatory cytokines expressed by local epithelial cells especially in ensuring the safety of sperm and to prevent infections – there is no reference

References: 1) Vighi, G.; Marcucci, F.; Sensi, L.; Di Cara, G.; Frati. F. Allergy and the gastrointestinal system. Clin Exp Immunol. 2008 Sep;153 487 Suppl 1(Suppl 1):3-6. doi: 10.1111/j.1365- 2249.2008.03713.x. PMID: 18721321; PMC ID: PMC2515351; 2) Baradaran Ghavami, S.; Pourhamzeh, M.; Farmani, M.; Raftar, S.K.A.; Shahrokh, S. Et al.. Cross-talk between immune system and microbiota in COVID-19. Expert Rev Gastroenterol Hepatol. 2021 Nov;15(11):1281-1294. doi:10.1080/17474124.2021.1991311 . 485 Epub 2021 Nov 2. PMID: 34654347; PMC ID: PMC8567289.- . – is nor relevant and does not relate to the topic

The chapter «Modern diet and lactobacillus issues» is not related to the topic and is not relevant.

P10L381 - diet habits (high sugar, refined food, cigarettes smokers) – this conclusion is not supported by arguments from the paper.

P10L386 (10) – «Nevertheless, screening and prevention still serve to base the formulation of diagnostic and screening measures for these infections in female population» - this conclusion also does not discuss in the main part of the paper.

Author Response

REVIEWER 1 Comments and Suggestions for Authors

P2L59 - this sentence is confused because it seem to mean that C,trachomatis is a high risk factor for cervical cancer development. In addition, there is no reference here.Thanks a lot for the valuable comment. The C Trachomatis is actually considered a co-risk factor together with HPV for insurgence of CC. Reference has been added.

Safaeian M et al. Chlamydia trachomatis e rischio di premalignità cervicale prevalente e incidente in una coorte basata sulla popolazione. J Natl Cancer Inst. 1 dicembre 2010;102(23):1794-804. doi: 10.1093/jnci/djq436. Epub 2010 Nov 23. PMID: 21098758; PMCID: PMC2994864.

P8L285 - bacterial/viral vaginosis (BVV) is a confused term and also there is no reference.

Different authors describe the double vaginosis infection using the acronyms BVV as mentioned in the text...

Thanks a lot for the valuable comment. The outcomes on female genital tract either at vaginal or cervical microbiota together with a substantial cytokine profiles has identified changes in microbiota diversity among women affected by HPV and C trachomatis infection, with bacteria/viral vaginosis (BVV) [27-30]

P2L47 - nearly 10% of the total newly diagnosed cancer cases 46 and 8% of the total cancer deaths - there is no appropriate reference.

Thanks a lot for the valuable comment. Cervical cancer (CC), is one of the most frequent cancer with one of the highest rate death among women worldwide, it counts nearly 10% of the total newly diagnosed cancer cases and 8% of the total cancer deaths [1].

P2L55 - The worldwide outcomes generally agree that top three C.trachomatis genotypes were The most common genotypes were E, followed by J, F, D, K, G, H, B, Ia. – there is no reference

Thanks a lot for the valuable comment. The worldwide outcomes generally agree that top three C. trachomatis genotypes were E, followed by J, F, D, K, G, H, B, Ia. It is generally accepted that H genotype is seen the closest with abnormal cervical cytology [3]. HPV and C trachomatis share common structural traits and both pathogen's DNA can be detected in around 99% of CC cases although the majority of patients remain asymptomatic or showing transient positivity with only a small number which eventually progress to cervical changes [3-5 ].

P3L90 - SNPs and low levels of immune-modulatory cytokines such as interferon gamma (IFN-γ), tissue growth factor beta (TGF-β) and interleukin 10 and 1β (IL-10, IL- 91 1β) – there is no appropriate reference.

Thanks a lot for the valuable comment. We have added the following:

Alagarasu, K.; Kaushal, H.; Shinde, P.; Kakade, M.; Chaudhary, U.; Padbidri, V.; Sangle, S.A.; Salvi, S.; Bavdekar, A.R.; D'costa, P.; Choudhary, M.L. TNFA and IL10 Polymorphisms and IL-6 and IL-10 Levels Influence Disease Severity in Influenza A(H1N1)pdm09 Virus Infected Patients. Genes 2021, 12, 1914. https://doi.org/10.3390/genes12121914.

P4L122 - Patients with both colorectal cancer and CC have showed lower levels of butyrate producers in the local microbiota than healthy individuals indicating an important shift in terms o local flora homeostasis [13-15]. –references 13-15 are not appropriate

Thanks a lot for the valuable comment. In fact, patients with both colorectal cancer and CC have showed lower levels of butyrate producers in the local microbiota than healthy individuals indicating an important shift in terms o local flora homeostasis [17-19].

19 Al-Qadami, G.H.; Secombe, K.R.; Subramaniam, C.B.; Wardill, H.R.; Bowen, J.M. Gut Microbiota-Derived Short-Chain Fatty Acids: Impact on Cancer Treatment Response and Toxicities. Microorganisms 2022, 10, 2048. https://doi.org/10.3390/microorganisms10102048

P5L151 - Changes in the vaginal microbiota composition across cervical carcinogenesis may also lead to a profound switch into high pH with ROS elevation, considered general markers indicative of inflammation, abnormal apoptosis, angiogenesis, hormonal imbalance, metabolic dysregulation and suggestive of suppressed local antitumor immunity and disease progression –reference 19 is not appropriate.

Thanks a lot for the valuable comment. The references are 23, 24 and 25 there is no 19

Gargiulo Isacco, C.; Ballini, A.; De Vito, D.; Nguyen, C.D.K.; Cantore, S. et al.. Rebalancing the Oral Microbiota as an Efficient Tool in Endocrine, Metabolic and Immune Disorders. Endocr Metab Immune Disord Drug Targets. 2021;21(5):777-784. doi:10.2174/1871530320666200729142504. PMID: 32727337.

Ayivi, R.D.; Gyawali, R.; Krastanov, A.; Aljaloud, S.O.; Worku, M.; Tahergorabi, R.; Silva, R.C.d.; Ibrahim, S.A. Lactic Acid Bacteria: Food Safety and Human Health Applications. Dairy 2020, 1, 202-232. https://doi.org/10.3390/dairy1030015.

Zhou, C.; Tuong, Z.,K.; Frazer, I.H. Papillomavirus Immune Evasion Strategies Target the Infected Cell and the Local Immune System. Front Oncol. 2019 Aug 2;9:682. doi: 10.3389/fonc.2019.00682. PMID: 31428574; PMC ID: PMC6688195.

P6L180 - The immanent problems are to be related to the simultaneously presence of multiple factors such as, high inflammatory state, the presence of single nucleotide polymorphisms (SNPs) on genes regulating the expression of immunosuppressive cytokines and interleukins like TGF- ß1, IL-10 and IFN-γ which favor local dysbiosis that in turn facilitate the persistence of the HPV and Chlamydia infection – there is no reference

Thanks a lot for the valuable comment. The immanent problems recall the simultaneously presence of multiple factors such as, high inflammatory state, the presence of SNPs on genes regulating the expression of immunosuppressive cytokines and interleukins and persistent local dysbiosis that in turn facilitate the long term aggressiveness of the HPV and Chlamydia infection [19,27].

P6L188 - The vaginal microbial ecosystem and the cytokine profile play a role in promoting 188 cervical dysplasia, given that an abnormal vaginal microbiota has been associated with 189 the acquisition of long term C trachomatis and HPV infection [25,26]. - references 25-26 are not appropriate

Thanks a lot for the valuable comment. We have changed the sentence and of course the references:

The co-infection with C trachomatis and HPV (especially the 16,18, 31,33, 53 and 56 genotypes) has been considered the most important risk factor for the presence of CC especially in young, unmarried women who started their sex lives early with several sexual partners and who used oral contraceptives. Interesting from these studies it comes out that prevalence of C trachomatis was high prevalently in young women [14-16].

P7L268 - In this scenario hormones are key actors as they are involved in regulating the production of antimicrobial peptides (β α-defensins, SLPI) and of pro-inflammatory cytokines expressed by local epithelial cells especially in ensuring the safety of sperm and to prevent infections – there is no reference

References: 1) Vighi, G.; Marcucci, F.; Sensi, L.; Di Cara, G.; Frati. F. Allergy and the gastrointestinal system. Clin Exp Immunol. 2008 Sep;153 487 Suppl 1(Suppl 1):3-6. doi: 10.1111/j.1365- 2249.2008.03713.x. PMID: 18721321; PMC ID: PMC2515351; 2) Baradaran Ghavami, S.; Pourhamzeh, M.; Farmani, M.; Raftar, S.K.A.; Shahrokh, S. Et al.. Cross-talk between immune system and microbiota in COVID-19. Expert Rev Gastroenterol Hepatol. 2021 Nov;15(11):1281-1294. doi:10.1080/17474124.2021.1991311 . 485 Epub 2021 Nov 2. PMID: 34654347; PMC ID: PMC8567289.- . – is nor relevant and does not relate to the topic

Thanks a lot for the valuable comment. In this scenario hormones play a fundamental role as they are involved in regulating the production of antimicrobial peptides (β α-defensins, SLPI) and of pro-inflammatory cytokines expressed by local epithelial cells especially in ensuring the safety of sperm and to prevent infections [35,36].

The references have been changed

Vighi et alt and Baradaran et al were changed with Siba, I.P.; Martynhak, B.J.; Pereira, M. When Gut Hormones Influence Brain Function in Depression. Appl. Biosci. 2023, 2, 31-51. https://doi.org/10.3390/applbiosci2010005

The chapter «Modern diet and lactobacillus issues» is not related to the topic and is not relevant.

We really appreciated the Reviewer’s comments however, we firmly believe that dysbiosis due to food and drink habits play an important role in immunity and in preserving the microbiota stability and therefore a a valuable screen against pathogens intrusion, that is why we added this chapter.

P10L381 - diet habits (high sugar, refined food, cigarettes smokers) – this conclusion is not supported by arguments from the paper.

Thanks a lot for the valuable comment. We have supported these statements by adding more information in the same paragraph and chapter. The modern industrialized "Unhealthy diet" habits are defined by high loading of sugar, solid oils, high refined carbohydrates and refined grains, fried potatoes and sweet drinks and, have been substantially associated with a higher BVV odds. For instance, overweight women are more prone to have higher rate if infections either bacterial or viral compared to lean women, thanks to different mechanisms including alterations in hormonal, metabolic, or immunological functions which tend to affect the overall structural stability of vaginal microbiota. For instance, due to high glycogen levels following high starch intake there was seen an altered condition of the composition of the genital fluids [39,40].

P10L386 (10) – «Nevertheless, screening and prevention still serve to base the formulation of diagnostic and screening measures for these infections in female population» - this conclusion also does not discuss in the main part of the paper.

Thanks a lot for the valuable comment. We have made the appropriate changes as follow in Conclusion section:

In summary, this manuscript strengthens the evidence individuals infected with both C. trachomatis and HPV have a heightened risk of developing cervical cancer. Therefore, it is necessary to expand C. trachomatis infection screening and proceed with early treatment, particularly among women at a higher risk of HPV infections. This approach will certainly raise the rate of protection against pelvic inflammatory disease and infertility, and potentially will be of great help in preventing and reducing the incidence of CC.

Reviewer 2 Report

Review Report

      The authors present three aspects related to the etiology of cervical cancer based on a literature review. However, not all aspects of the review, presented by the authors, support the objective and the conclusions of the study.     The text needs a thorough revision of the English language. The English is sometimes very difficult to understand.

Author Response

REVIEWER 2 Comments and Suggestions for Authors

Review Report

The authors present three aspects related to the etiology of cervical cancer based on a literature review. However, not all aspects of the review, presented by the authors, support the objective and the conclusions of the study. The text needs a thorough revision of the English language. The English is sometimes very difficult to understand.

Thanks a lot for the valuable comment. We have proceeded to a thorough revision of the English and we made valuable changes to support our position. Changes can be seen highlighted in yellow in the main text.

Reviewer 3 Report

I found repetitive tyext

page 2 line 60 and page 2 line 64

Page 3 line 94,95 and page 3 line 112,113. In this text what is the reference? Only reference 9 matches

Author Response

REVIEWER 3

I found repetitive tyext

page 2 line 60 and page 2 line 64

Page 3 line 94,95 and page 3 line 112,113.

In this text what is the reference? Only reference 9 matches

We thanks the reviewer helpful suggestions we have proceeded to the correct following the comments.

Round 2

Reviewer 2 Report

Alterations of Vaginal Microbiota and Chlamydia trachomatis as crucial co-causative factors in cervical cancer genesis procured by HPV 

Manuscript ID: microorganisms- 2139045

Type of manuscript: Review

Review Report 2

Dear MS  Hana Li     

      In the second presentation of the article, the authors not always presented the references that gave rise to the text, which weakens the review. Also, the text of the second presentation of the article needs a thorough revision of the English language.

    The attached review report has highlightings where references insertion is needed.

    My overall recommendation is to reconsider the acceptance of the review after a major revision. 

Author Response

Dear Reviewer

1 we appreciated your valuable comments and we have gone through every single request and suggestion especially in referring to the reference within the main text (each ref number is highlighted in yellow).

2 The conclusion section has been corrected

3 The whole manuscript went through consistent English grammar revision and correction

4 We have also added some references and of course, we have expanded the main text
